# Bridge Monitoring Strategies for Sustainable Development with Microwave Radar Interferometry

**Lilong Zou** [1] , **Weike Feng** [2] , **Olimpia Masci** [3], **Giovanni Nico** [4] , **Amir M. Alani** [5,*] **and Motoyuki Sato** [6]

1. School of Computing and Engineering, University of West London, London W5 5RF, UK; lilong.zou@uwl.ac.uk
2. Early Warning and Detection Department, Air Force Engineering University, Xi'an 710051, China; fengweike007@163.com
3. DIAN s.r.l., Via Ferruccio Parri, 44, 75100 Matera, Italy; o.masci@dianalysis.eu
4. Istituto per le Applicazioni del Calcolo (IAC), Consiglio Nazionale dele Ricerche (CNR), 70126 Bari, Italy; g.nico@ba.iac.cnr.it
5. Faculty of Engineering, Computing and the Environment, Kingston University, London KT1 1LQ, UK
6. Center for Northeast Asian Studies, Tohoku University, Sendai 980-8576, Japan; motoyuki.sato.b3@tohoku.ac.jp
* Correspondence: m.alani@kingston.ac.uk; Tel.: +44-208-417-5476

**Abstract:** The potential of a coherent microwave radar for infrastructure health monitoring has been investigated over the past decade. Microwave radar measuring based on interferometry processing is a non-invasive technique that can measure the line-of-sight (LOS) displacements of large infrastructure with sub-millimeter precision and provide the corresponding frequency spectrum. It has the capability to estimate infrastructure vibration simultaneously and remotely with high accuracy and repeatability, which serves the long-term serviceability of bridge structures within the context of the long-term sustainability of civil engineering infrastructure management. In this paper, we present three types of microwave radar systems employed to monitor the displacement of bridges in Japan and Italy. A technique that fuses polarimetric analysis and the interferometry technique for bridge monitoring is proposed. Monitoring results achieved with full polarimetric real aperture radar (RAR), step-frequency continuous-wave (SFCW)-based linear synthetic aperture, and multi-input multi-output (MIMO) array sensors are also presented. The results reveal bridge dynamic responses under different loading conditions, including wind, vehicular traffic, and passing trains, and show that microwave sensor interferometry can be utilized to monitor the dynamics of bridge structures with unprecedented spatial and temporal resolution. This paper demonstrates that microwave sensor interferometry with efficient, cost-effective, and non-destructive properties is a serious contender to employment as a sustainable infrastructure monitoring technology serving the sustainable development agenda.

**Keywords:** sustainable infrastructure monitoring; bridge monitoring; microwave radar; interferometry; polarimetry; sustainable development

## 1. Introduction

Most constructed structures, including buildings, roadways, and bridges, require regular inspection and assessment so that appropriate action can be taken to maintain their effectiveness. This action could be in the form of Principal Bridge Inspections or general inspections, which are conducted annually and monitored by the Highways Agency. The cost of correcting defects in buildings and civil infrastructure is immense in many countries [1]. A study carried out in the U.S. estimated that the failure of civil infrastructure systems to perform at their expected level can reduce the national gross domestic product, while a study conducted within the UK showed the immense annual cost of correcting defects in buildings and civil engineering structures—a sum greater than the total profits of

all UK construction companies [2]. For this reason, in many developed countries, where numerous infrastructure systems have become old and deteriorated, the need for efficient maintenance strategies has become crucial. Thus, on the one hand, the maintenance of highways and associated structures, such as bridges, is a pressing issue, while on the other hand, local authorities who own infrastructure are reluctant to carry out costly maintenance, such as strengthening works [3]. Making good maintenance decisions is, therefore, important, but it relies on the application of years of practical experience.

Traditionally, visualization inspections on-site have been the method for assessing bridge conditions [4]. However, this approach has limitations. It detects visible damage only, overlooks hidden problems before they surface, and relies heavily on the inspector's experience and judgment. On-site visual assessments are neither economical nor appropriate for ongoing monitoring [5]. Infrastructure health monitoring has been shown to be quite successful in the past several years in the routine evaluation of structural problems, and it is now more accessible—especially for bridges—due to lower data gathering costs [6]. By proactively identifying possible issues and estimating the remaining lifetime, this method highlights how crucial it is to comprehend the existing condition and load-bearing capability for long-term bridge health. Damage to a bridge's construction that results from explosions or natural catastrophes, such as earthquakes, affects the material's characteristics, boundary conditions, and structural integrity [7]. Structural problems are also a result of service load factors, including aging, traffic expansion, increasing degradation, and environmental effects [8,9]. The benefit of putting in place a bridge health monitoring system is that it allows for an objective evaluation of the structure's state over time. This proactive strategy makes it possible to identify damage in the aftermath of a catastrophic occurrence or before it becomes a serious problem [10].

Various methodologies exist for monitoring the health of bridges at different stages, with a prominent approach being vibration-based testing. This method relies on dynamic testing to detect structural changes, assuming that damage affects stiffness rather than mass loss [11–13]. Bridge health may be effectively monitored and detected through the use of dynamic testing, which assesses the mechanical reaction of a structure's deformation. This method has attracted a lot of research interest. Dynamic testing techniques frequently include piezoelectric accelerometers and fiber optic sensors, which enable the precise and dependable recording of dynamic deformation time series [14–16]. However, the precise placement of sensors in predetermined places and the required hardwiring to the data gathering system are prerequisites for deploying these techniques on monitored bridges. Although this method works well, it is expensive, time-consuming, and there is a chance that ancient bridges will be harmed. Balancing the benefits of dynamic testing with the challenges of installation is crucial for optimizing bridge health monitoring strategies [17–21].

Ground-based real aperture radar (GB-RAR) systems and space-borne synthetic aperture radar (SAR) systems are the two primary types of microwave interferometric radar systems, which are essential for a variety of applications. Because of its ability to provide wide-area and semi-continuous monitoring, space-borne interferometry SAR (InSAR) is a popular option for a number of applications [22–25]. Both ground-based and satellite radar interferometers share fundamental principles, differing only in viewing geometry. Interferometric radar relies on the phase information of back-reflected microwave signals to detect target displacement within the target cone. However, measuring the distance directly from the phase information encounters challenges, including ambiguity phases and sensitivity to atmospheric conditions and sensor position changes. The electrical device's capacity to detect minute phase rotations and carry out further signal processing determines how accurate the interferometry phase will be. In a perfect world, 0.1 mm precision could be attained with electromagnetic waves in the Ku band [26–28]. Space-borne InSAR measures provide weekly updates and great spatial resolution, enabling millimeter-level precision in tracking geometrical changes. Estimating these components from multitrack radar data is made possible by infrastructure deformation, particularly in vertical and horizontal

directions. On the other hand, the track direction affects the sensitivity. Balancing accuracy and various factors in radar systems is thus critical for optimizing their effectiveness in monitoring and detecting structural changes.

The GB-RAR interferometry method is a widely embraced technology for measuring the dynamic deformation of bridges [29–36]. This methodology delivers accurate sub-millimeter deformation monitoring in a non-contact manner. Notably, it offers a straightforward and rapid setup for investigations while enabling a detailed assessment of the condition of the entire bridge structure. In this paper, we focus on the use of ground-based SAR (GB-SAR) interferometric sensors, since they offer flexible application and greater degrees of freedom in terms of geometric view and temporal baselines. The GB-SAR technique has reached maturity in the deformation monitoring of landslide-prone areas and dams, both for periodic measurement and for implementation with continuously operating systems.

Research within the field of GB-SAR data processing remains ongoing, especially for the robust estimation of displacements and vibration frequencies and the development of sensor networks and methodologies for data integration, potentially offering the opportunity to analyze different observations in a spatial and temporal context. In this research work, we describe the application of GB-SAR interferometry to the monitoring of bridges. Three case studies applying different types of microwave radar systems to cases relating to a metallic bridge across a motorway, a stone bridge, and a concrete bridge crossing a river are discussed. The structure of the paper is as follows. Section 2 introduces the basic principle of GB-SAR interferometry for the measurement of aerial displacements and vibration frequencies. Section 3 summarizes the GB-SAR results for the three case studies. Section 4 draws some conclusions and suggests new research directions for the InSAR monitoring of bridges.

## 2. Methodology

### 2.1. Microwave Radar Interferometery

The space-borne synthetic aperture radar system and the ground-based synthetic aperture radar system are the two primary types of microwave interferometric radar systems that have been used for specific purposes in recent years. These systems are categorized based on their respective radar system platforms. Wide-area, semi-continuous monitoring may be effectively accomplished with space-borne interferometry SAR. With a high spatial resolution and biweekly data update, the radar measurements are able to identify variations in the track geometry to within millimeters. Multitrack radar data may be used to estimate the two components of the deformation vector when the infrastructure's deformation is mostly restricted to the vertical and horizontal transversal directions, albeit its sensitivity varies depending on the track's orientation.

Phase measurements from coherent scatterers are necessary for deformation monitoring with InSAR. These observations may be double-differenced to compare phase differences between two sites and two epochs. By multiplying the phase differences by the radar wavelength (usually in the order of cm) and accounting for the radar pulse's two-way trip, differential displacements may be obtained from this [37]. Buildings, bare rocks, and railroads are examples of targets with consistent phase values in time that represent coherent scatterers. For a given ground target, temporal behavior is described as the deformation time series with multiple differential displacements at the sequential epochs [38–40].

The systematic phase from the flat term $\Delta\phi_{flat}$, the topographic phase term $\Delta\phi_{topo}$, the deformation phase term $\Delta\phi_{def}$, the ionospheric phase term $\Delta\phi_{ion}$, and the phase noise term $\Delta\phi_{noise}$ combine to generate the differential interferometric phase in satellite SAR interferometry. Hence, the interferometry phase in satellite SAR can be written as:

$$\Delta\phi = \Delta\phi_{flat} + \Delta\phi_{topo} + \Delta\phi_{def} + \Delta\phi_{ion} + \Delta\phi_{noise} + 2n\pi, \tag{1}$$

where $2n\pi$ indicates the phase wrapping term. The displacement phase in the radar LOS direction may be derived once the non-deformation-caused phase components have been removed from the interferogram. Only the projection of the (3D) deformation phase $\Delta\phi$ onto the radar LOS $\Delta\phi_{LOS}$ is detected by radar measurements. The projection is specified as $\Delta\phi_{geo} = [\Delta\phi_e; \Delta\phi_n; \Delta\phi_u]^T$ in practice, and it contains the deformation components in the east, north, and up directions, respectively. $\Delta\phi_{LOS} = P^T \Delta\phi_{geo}$, where $P$ is the projection vector, may be used to apply the projection:

$$P = [-\sin\theta\cos\alpha_h, \sin\theta\sin\alpha_h, \cos\theta]^T, \tag{2}$$

where $\theta$ is the local incidence angle and $\alpha_h$ is the satellite track's instantaneous heading (azimuth with respect to north).

The calculation of 3D deformation components based on LOS measurements is ill-posed with one satellite and one viewing geometry, resulting in three unknowns versus one observation. Thus, until additional limitations can be applied, 3D deformation cannot be determined from a single satellite. A more thorough explanation of this subject can be found in [41–43].

*2.2. Ground-Based Radar Interferometry*

The only distinction between satellite-based and ground-based radar interferometers is in their viewing geometry. Ground interferometric radar is a type of sensor that uses the phase information of the microwave signal that is back-reflected to determine the target's movement inside the target cone. The phase difference of the rearward reflected wave varies on the distance between the radar and the target if only the single-point scattering of monochromatic wave impact is taken into account. Nevertheless, since this phase has to be impacted by an ambiguity phase that is equivalent to half a wavelength, it cannot be directly used for measuring the distance. Additionally, variations in the sensor's location and the current state of the air have an impact. Therefore, the interference phase $\Delta\phi_{21}$ obtained at two moments can be expressed as [44]:

$$\Delta\phi_{21} = \Delta\phi_{def} + \Delta\phi_{atmo} + \Delta\phi_{geo} + \Delta\phi_{noise} + 2n\pi, \tag{3}$$

where $\Delta\phi_{def}$ is the component related to the displacement, $\Delta\phi_{atmo}$ is the phase component due to the atmospheric effects during image acquisition, $\Delta\varphi geo$ contains the geometric phase component due to repositioning errors between the two acquisitions, $\Delta\varphi noise$ is the phase noise component and, finally, the term $2n\pi$ arises due to phase wrapping that corresponds to the half-wavelength ambiguity phase. Equation (3) is similar to Equation (1) used for satellite SAR interferometry observations. Therefore, interferometry phase accuracy is dependent on the signal processing that comes after, in addition to the electronic device's capacity to recognize minor phase rotation. In an ideal situation, by using electromagnetic waves in the Ku band, accuracy can reach 0.1 mm, which is much higher than that of satellite SAR interferometry observations.

When using a real aperture scenario, the radar sends out a modulated signal that enables the sensor to detect displacements in objects that are farther apart than the radar's resolution. Range-only resolution monitoring is insufficient in many situations. To offer angular resolution as well, the microwave radar system must be rotated or moved. By doing this, a 2D picture that allows for azimuth discrimination between objects at the same radar distance may be produced. To put it another way, instead of only seeing a plot of the displacement pattern's projection in the view direction, a 2D picture of the pattern may be obtained by moving the microwave radar system. Despite being a significant feature, this causes the sensor to become slower, heavier, and more clunky, which makes it a less frequently used alternative.

## 3. Bridge Monitoring

In this section, we describe the case studies monitoring different types of bridges with different microwave radar systems.

### 3.1. Polarimetric GB-RAR System

Using a frequency-modulated continuous-wave (FMCW) radar operating in the 17–17.3 GHz band with a 0.5 m range resolution, the GB-SAR system was employed in this study. The radar system is completely polarimetric and has four horn antennas. These antennas are divided into two pairs: one for broadcasting electromagnetic waves with polarization in both directions, and the other pair for receiving waves with polarization in both directions. The radar device was set up on a tripod next to the bridge, with its antennae facing in the direction of the building, as shown in Figure 1a. The long-span metallic railway bridge is part of the Joban Line Railway, operated by the East Japan Railway Company. The bridge spans the Abukuma River, connecting Iwanuma City to Watari Town with a dual-lane railway. The system parameters are shown in Figure 1b. The measurements were obtained when train traffic was operating normally. This project aims to enable dynamic monitoring of the bridge during a train crossing. By evaluating the deformation scale, the bridge's overall health may be determined. With a little sample time, an FMCW radar system may provide a radar profile. We used 0.002 s as the sample time in this investigation. The profiles for the radar cross-section (RCS) of each polarimetric response of this bridge are displayed in Figure 2.

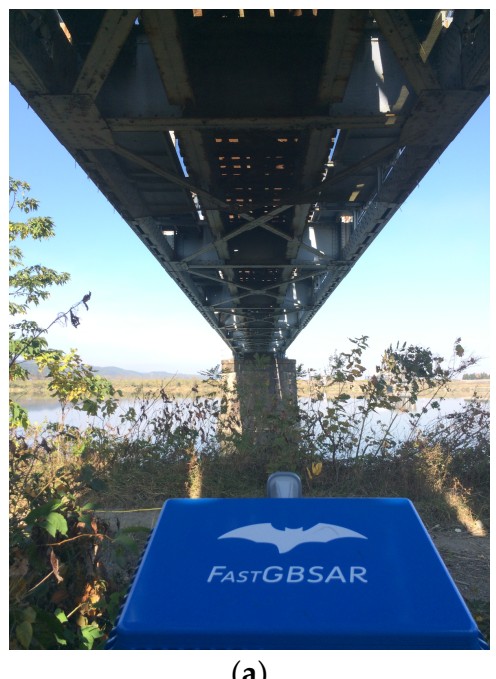

| Parameter | Value |
|---|---|
| Operating frequency | 17.2 GHz (Ku Band) |
| Range resolution | Up to 0.5 m |
| Maximum range | 4 km |
| EIRP power | 19 to 42 dBm |
| Operating temperature range | -25 $^o$C to 60 $^o$C |
| Sensor weight | 10 kg |
| Accuracy | ± 0.01 mm |
| Power consumption | 70 W |

(**a**)                                    (**b**)

**Figure 1.** (**a**) Polarimetric GB-RAR system placed beneath the bridge. (**b**) GB-RAR system parameters.

The acquisition lasted approximately 300 s. The RCS consistently drops with the increasing distance because of the acquisition geometry and the bridge's scattering characteristics. The detected bridge reaction starts at a range of 20 m. Figure 3 displays the appropriate temporal trends of the LOS displacement. Each profile was identified by the range distance of the target to which it refers. It can be observed that, between 180 s and 270 s, in each of the range profiles that match the objectives on the bridge, displacement was measured. Only a few of the structure's reflection sites were able to extract precise displacement time series due to the signal-to-noise ratio (SNR).

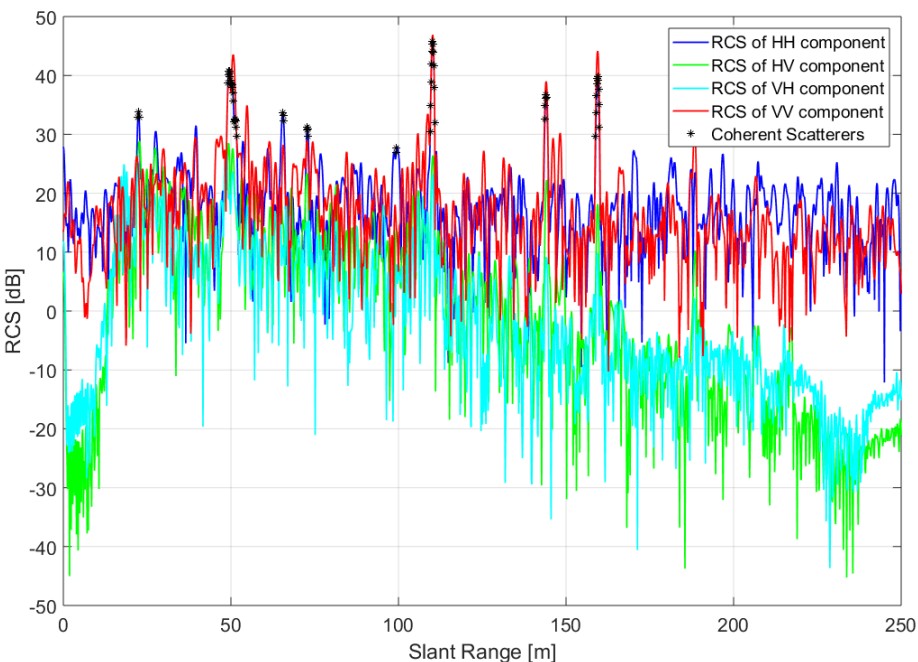

**Figure 2.** Radar cross-section (RCS) profiles of each polarimetric response.

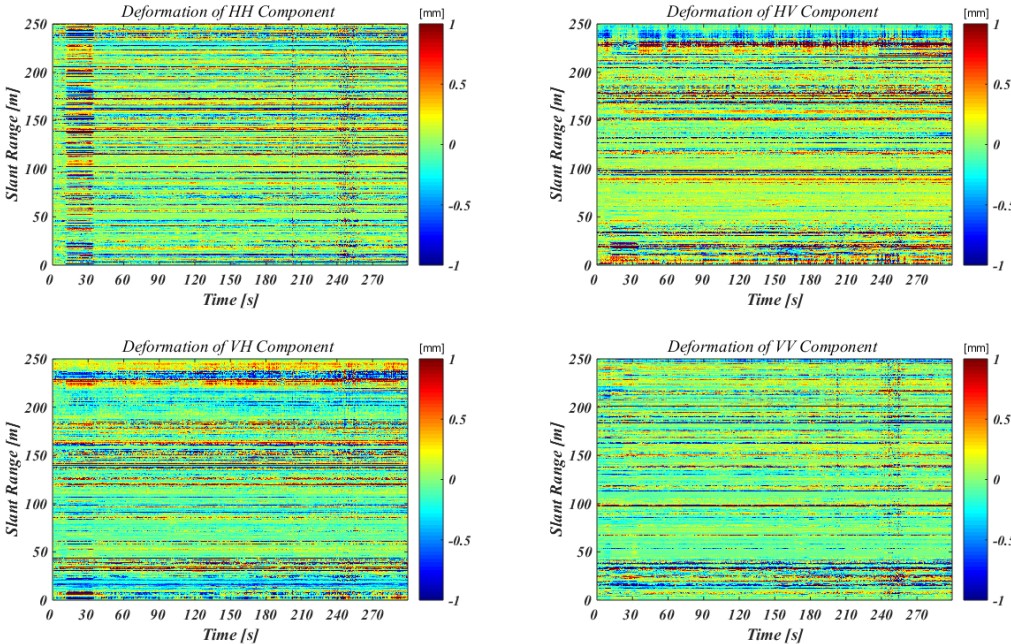

**Figure 3.** LOS displacement of each polarimetric response.

These high SNR reflecting points or stable points are known as coherent scatterers and can be selected based on the estimation of the dispersion of the index (DA), which is defined as [45,46]:

$$DA = \frac{A}{mA},$$ (4)

where mA and A are the mean value and standard deviation of the amplitude samples of the pixel along the waveform, respectively. Precise statistical hypotheses allow one to relate DA to the phase dispersion. When a point target is illuminated at different times, the amplitude of the signal backscattered from a stable point and affected by additive noise is rice distributed. In Figure 2, the black star indicates the chosen coherent scatterers. The real aperture radar detected no movement along the azimuth direction with a high

pulse repetition frequency (PRF) rate but at a cost: no azimuth resolution. All targets within the antenna beam with the same slant range were recorded together, making it difficult to identify the corresponding target from the radar diagram. One solution is a full polarimetric analysis. The full polarimetric responses hold the maximum information of the scatterers, and all the polarimetric information can be written into a scattering matrix. Several methods that were previously used to decompose the polarimetric scattering matrix are now used on parameter spaces that are phenomenologically more significant [47]. In this particular scenario, complete polarimetric information might be used for target identification and categorization based on these methods. For additional investigation, the absolute amplitude of these coherent scatterers was employed in Figure 4a. The absolute amplitude of the HH component is located on the horizontal axis, while the absolute amplitude of the VV component is located on the vertical axis. The coherent scatterers are dispersed down the horizontal axis, as this image shows. This indicates that the replies for the HH component were substantially bigger than those for the VV component. We inferred that the horizontal construction of the bridge was the source of these reflections because of the acquisition geometry. Moreover, H/Alpha analysis was carried out based on the full polarimetric responses of the coherent scatterers. The alpha angle may be used to determine average scattering processes, and entropy is a natural measure of the intrinsic reversibility of the scattering data. By taking various scattering behaviors into account, the H/Alpha plane may be separated into nine fundamental zones, which will help to segregate the data into a basic scattering mechanism [48]. The backscattered signal identifies dominant scattering processes, which are represented by the pixel indices in each zone, separately [49]. From the results of the H/Alpha analysis shown in Figure 4b, we can see that most of the coherent scatterers arose from the region with low entropy and high alpha values. This probably means that the reflections arose from similar dihedral reflector- or dipole-shaped targets. Based on this analysis, a dipole-like target (red dot in Figure 4a,b) was selected, which we stated would be the horizontal needle beam located 14.5 m from the system to the bridge. The displacement profile corresponding to this target located at the range distance R = 14.5 m is displayed in Figure 5. The maximum displacement value around 2.5 mm is shown. The negative displacement means that the displacement was toward the radar position. After the train crossed the bridge, small, damped oscillations were observed.

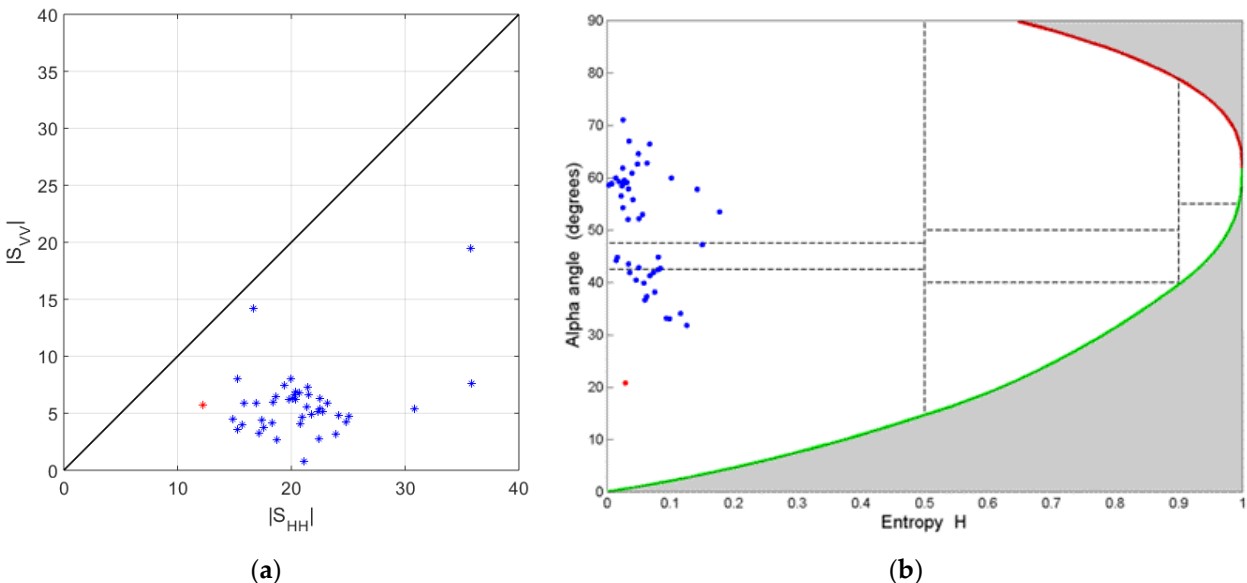

(**a**)                    (**b**)

**Figure 4.** (**a**) Polarimetric data distribution in HH and VV components of RCS. (**b**) Polarimetric data distribution of the bridge in the 2D H/Alpha plane. Red dot indicated the selected reflection point located at 14.5 m; Blue dot indicated others reflection points.

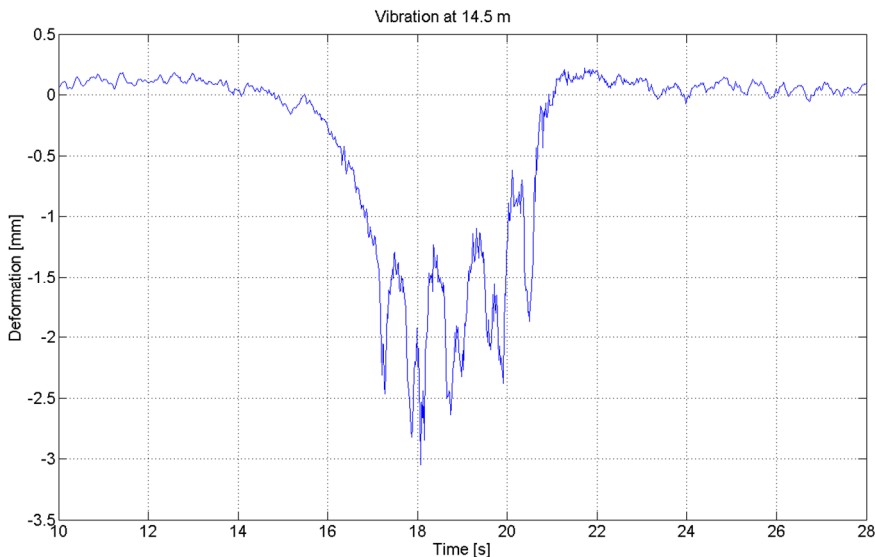

**Figure 5.** Vibration deformation during train passing located at 14.5 m.

*3.2. GB-SAR System*

In this section, two case studies are described. In the first case study, 1D radar profiles were used to study a metallic railway bridge across a motorway (San Severo, Italy). Usually, for this kind of application, corner reflectors (CRs) are installed on the bridge to materialize targets with a strong microwave reflectance, which is useful for both georeferencing the radar measurements with respect to the bridge and identifying key points on the structure from where to provide estimates of displacement and vibration frequencies. The metallic bridge is a railway bridge located close to San Severo (southern Italy). The bridge crosses the highway. The bridge span is about 30 m, with the two concrete pillars located on the two sides of the highway, as shown in Figure 6. The radar was installed close to one of the pillars of the bridge, looking toward the bridge platform, and RAR was acquired. The aim of the study was to provide dynamic monitoring of the bridge when being crossed by a train. The sampling time of the radar profiles was $\Delta t = 0.0109$ s. Figure 7 shows the radar cross-section (RCS) profile acquired at 19 h, 12 m, and 40 s. The acquisition lasted approximately 60 s. The RCS consistently reduced as the range distance grew because of the bridge's scattering characteristics and the acquisition geometry. A range of 15 to 90 m allows for the identification of the bridge reaction. Figure 8 displays the appropriate temporal characteristics of the LOS displacements. The target's range distance serves as a unique identifier for each profile. It is evident that displacement was detected in all range profiles corresponding to targets on the bridge, during the time interval of 30 to 40 s. Displacement was clearer for targets located up to R = 70 m from the radar but could also be recognized for targets located beyond this range, even where noisy strips appeared, probably due to the measurement configuration and some multi-trajectory scattering responses. The profiles corresponding to targets located at ranges of R = 21 m, 21.75 m, 22.5 m, and 23.25 m are displayed in Figure 9. A displacement value between D = −1.5 mm and D = −2.5 mm is shown. A negative displacement means that the displacement was located toward the radar position. After the train crossed the bridge, damp oscillations were observed.

Figure 10 provides more detailed information on the profile measured at range R = 21 m when the train was crossing the bridge. A sequence of peaks was observed after t = 35 s, probably related to the passage of the different train coaches. At the bottom of Figure 10, the power spectrum of bridge displacements is reported. Two peaks could be observed at frequencies of $f_1 = 2.4$ Hz and $f_2 = 4.2$ Hz.

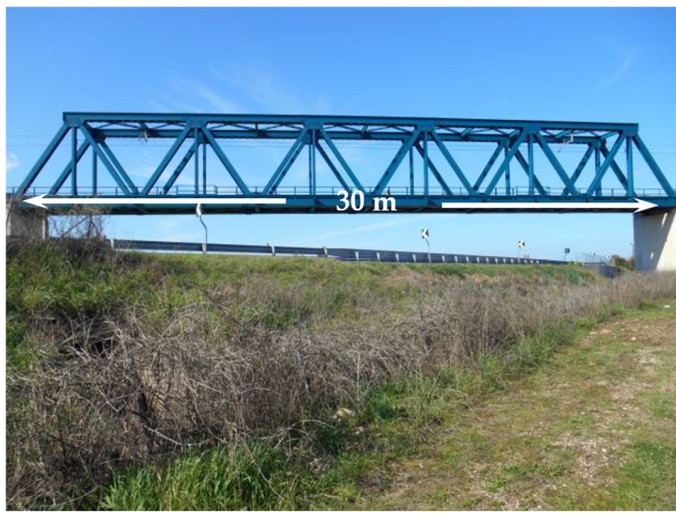

**Figure 6.** Metallic railway bridge across a motorway (San Severo, Italy).

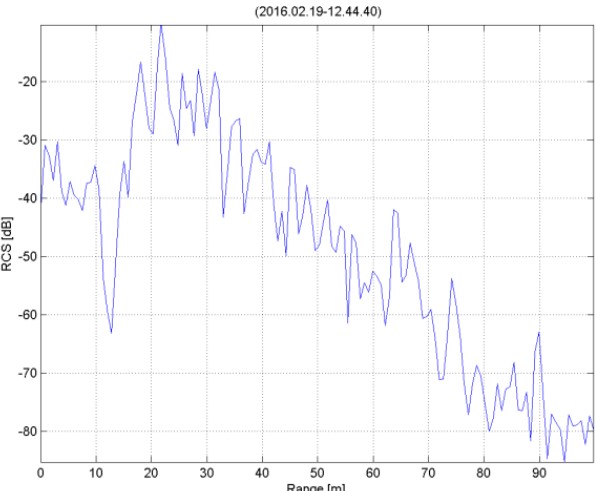

**Figure 7.** Radar cross-section (RCS) profile.

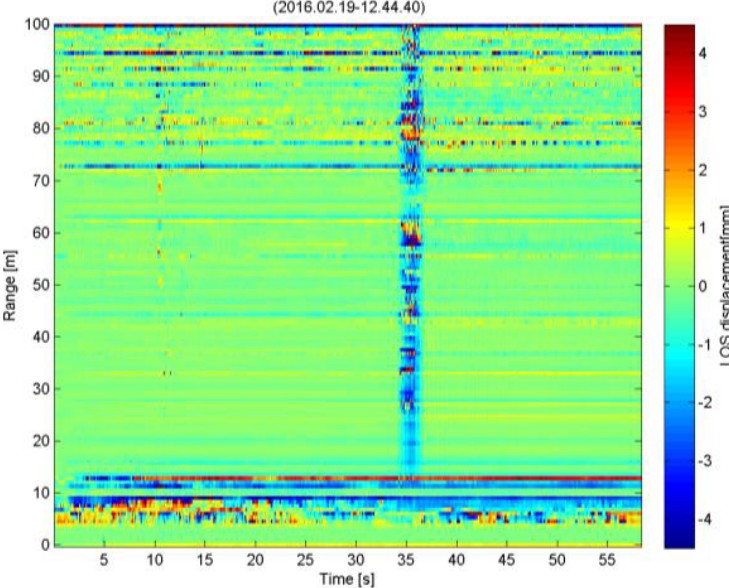

**Figure 8.** Temporal profiles of LOS displacements.

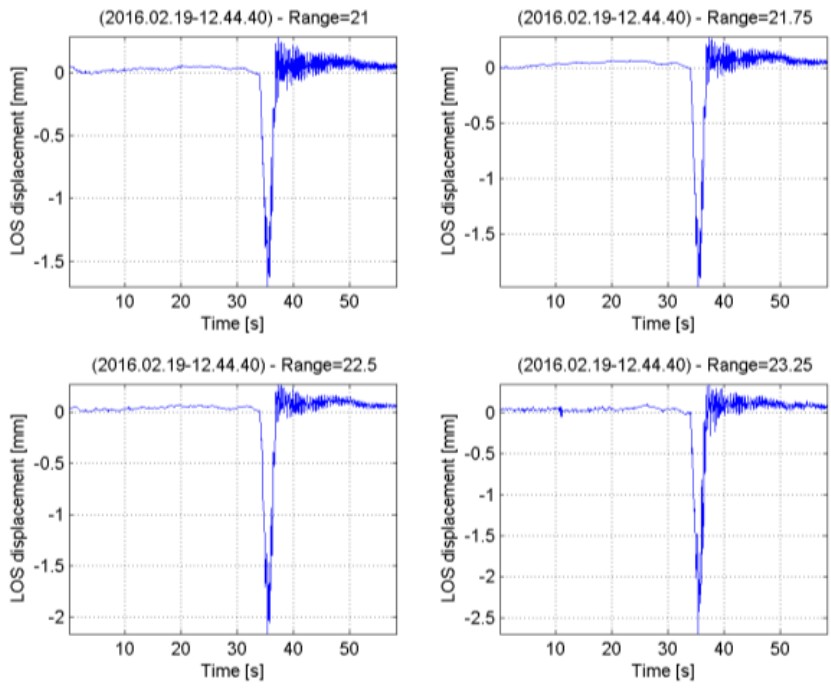

**Figure 9.** Temporal profiles of LOS displacements at different ranges.

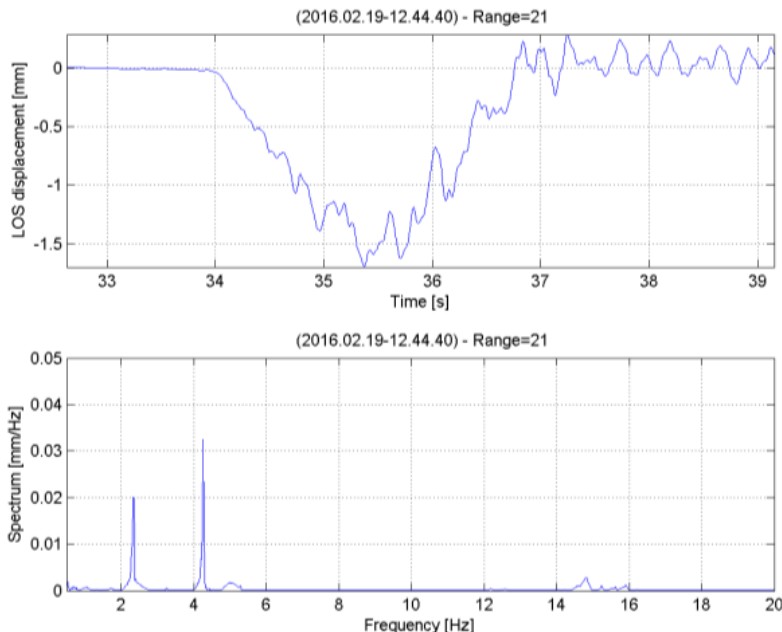

**Figure 10.** LOS displacements and corresponding frequencies.

In the second case study, 2D SAR images were used to measure the LOS displacements of a stone bridge across the Basento River (Grassano, Italy), as shown in Figure 11. The aim of this study was to measure differential displacements of the stone bridge possibly related to river action. Such displacement measurements could be useful when assessing hydraulic hazards corresponding to bridges crossing rivers. The stone bridge is located close to Grassano (southern Italy). It is a road bridge crossing the Basento river. It consists of multiple spans, each of a length smaller than 8 m. Figure 12 displays the normalized RCS (NRCS) image of the bridge and the surrounding area. The radar has been installed in front of the bridge on a concrete platform not affected by the river's temporal changes. The stone bridge and the metallic structure located between the radar and the bridge were mapped in the range/azimuth plane as two almost parallel curves separated by a range

distance of approximately 20 m. The pixels corresponding to the bridge signature in the NCRS images started on the left at an angular position of about $40^{\circ}$ and a range distance of $R = 32$ m, and continued along the curve up to an angular position of approximately $20^{\circ}$ and a range distance of $R = 93$ m. The NCRS image in Figure 12 also shows part of the river gravel in front of the radar up to range distances of around 30 m. Three measurement campaigns were conducted on 8 June 2015, 28 August 2015, and 28 October 2015, acquiring 20 SAR images during each campaign. The temporal baseline of SAR acquisitions in each measurement campaign was 5 m and 28 s. The maps reported in Figure 13 show the displacements of the bridge on a temporal scale of two months and four and a half months, respectively. The bridge corresponds to the upper curve reported in the two maps.

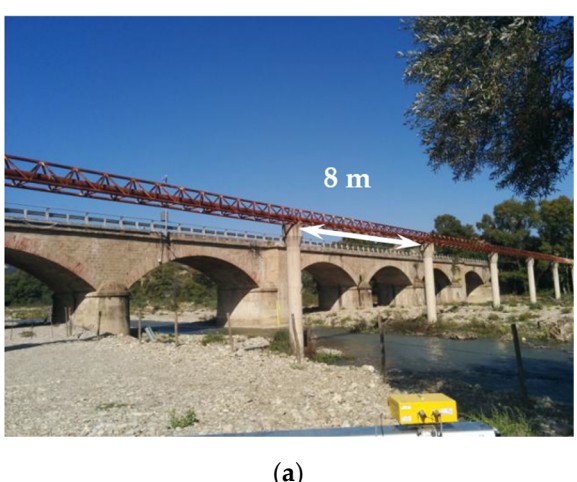

| Parameter | Value |
|---|---|
| Operating frequency | 17.2 GHz (Ku Band) |
| Range Resolution | Up to 0.5 m |
| Maximum Range | 4.5 km |
| Central Wavelength | 17.44 mm |
| Scan Length | 2 m |
| Scan time | 5 min |
| Cross-Range Resolution | 4.4 mrad |
| Accuracy | ± 0.02 mm |

(**a**)            (**b**)

**Figure 11.** (**a**) Stone bridge across Basento River (Grassano, Italy). (**b**) LOS displacements measured at the stone bridge between 8 June and 28 October 2015.

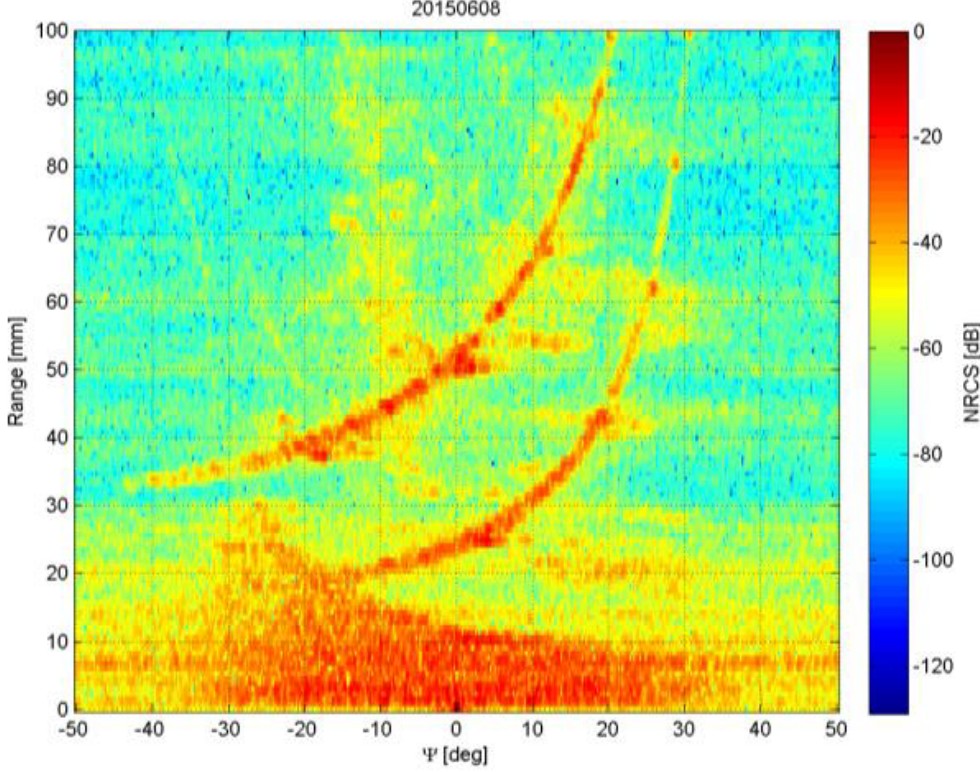

**Figure 12.** NRCS image of the stone bridge across Basento River (Grassano, Italy).

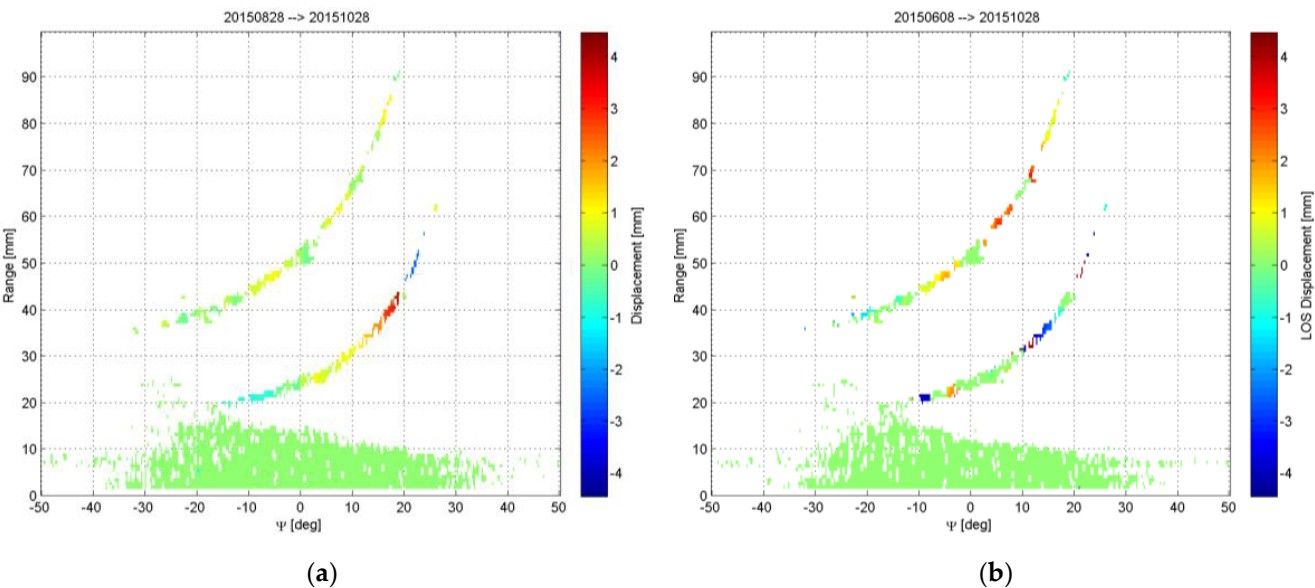

**Figure 13.** (**a**) LOS displacements measured at the stone bridge between 28 August and 28 October 2015. (**b**) LOS displacements measured at the stone bridge between 8 June and 28 October 2015.

The large stone pillars located in the river and those close to the riverbank appeared stable in both the displacement maps. In contrast, displacements in the corresponding bridge arches were measured. Displacements of 1 mm and 3 mm were measured between 28 August and 28 October 2015 and between 8 June and 28 October 2015, respectively. A positive displacement indicates movement away from the radar. It is worth noting that measured displacements corresponding to the lower curve probably contain only the component caused by the thermal variations of the metallic girder on the top of the concrete pillars, located between the radar location and the bridge (see Figure 11). The bridge was also studied using a RAR configuration, with the radar installed below one of the bridge arches and the TX/RX antennas pointed almost vertically toward the bridge, to measure possible vertical deformation when a truck crosses the bridge. Figure 14 shows the RCS profile of RAR measurement. The peak in the radar response was at around $R = 9$ m. The LOS displacement profile and the corresponding frequency spectrum measured at this range distance are reported in Figure 15. It can be observed that the maximum vertical displacement of the bridge was about 0.2 mm from the radar and not toward the radar, as expected. Furthermore, the frequency spectrum showed no frequency peak.

*3.3. MIMO Array Radar Interferometry*

The developed SFCW-based MIMO array radar system is shown in Figure 16. The system includes eight spiral antennas as transmitters and eight Vivaldi antennas as receivers. Its working frequency ranges from 3.81 to 8.01 GHz with 1024 steps (adjustable; the minimal number is 8), resulting in a range resolution of 3.53 cm and a maximal unambiguous range of about 36 m. The length of its equivalent uniform array is approximately 0.89 m, yielding an angle resolution of 0.028 rad. The data sampling rate of this system is about 8 Hz, which is faster than that of commercial GB-SAR systems. More detailed information on this MIMO radar system can be found in [50]. For the designed MIMO array radar system, the time division multiplexing (TDM) approach was used to achieve transmission orthogonality (i.e., each transmitter transmits the same SFCW signal sequentially using a switch network, and a delay compensation method was used to reduce the negative influence of time synchronization error). More detailed information relating to this MIMO radar system can be found in [50]. The signal processing flowchart for the designed MIMO radar system is shown in Figure 17.

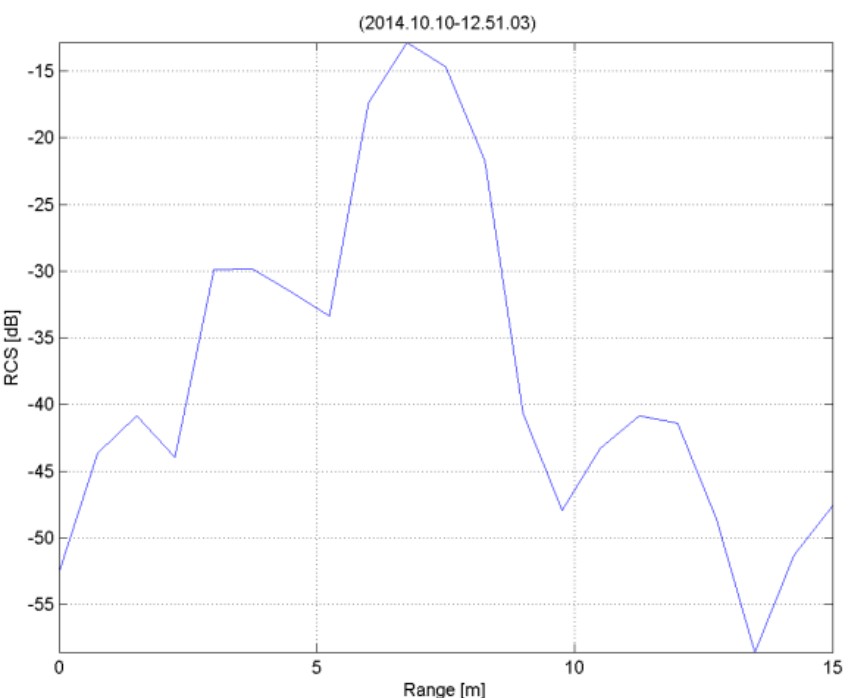

**Figure 14.** RCS profile acquired at the stone bridge.

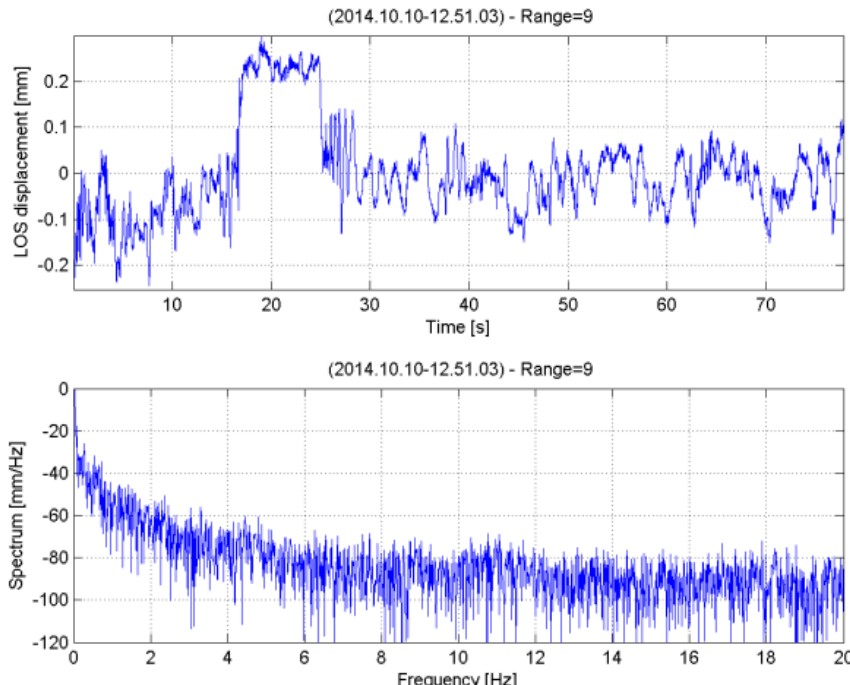

**Figure 15.** LOS displacement profile measured at the stone bridge at a range of R = 9 m while a truck was crossing (**above**), and the corresponding frequency spectrum (**below**).

Setting three trihedral corner reflectors (CRs) as the targets, the MIMO radar imaging result obtained by the back projection (BP) algorithm is shown in Figure 18. It can be seen that three CRs could be well focused and distinguished. Further, we mounted a CR on a linear scanner that can be controlled with an accuracy of 1 μm, as shown in the left figure of Figure 18. Along the LOS direction, the CR was moved away from the designed MIMO radar system, from 2 to 22 mm, with a 2 mm step. The displacement estimation result is shown in the right figure of Figure 19, where the root mean square error of the estimated

displacement was about 0.37 mm, which proves the feasibility of the developed MIMO array radar system.

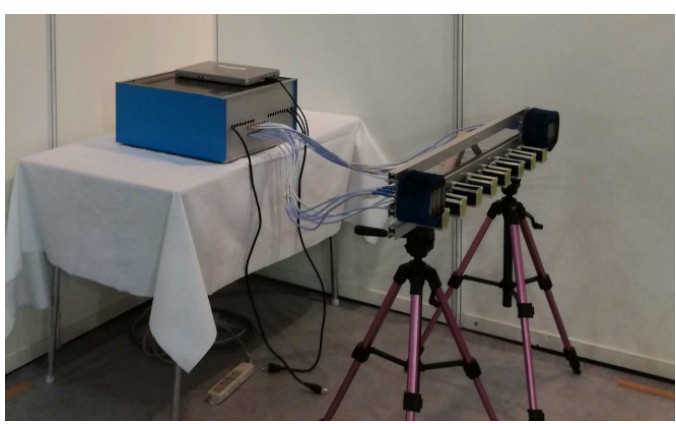

**Figure 16.** Designed MIMO radar system.

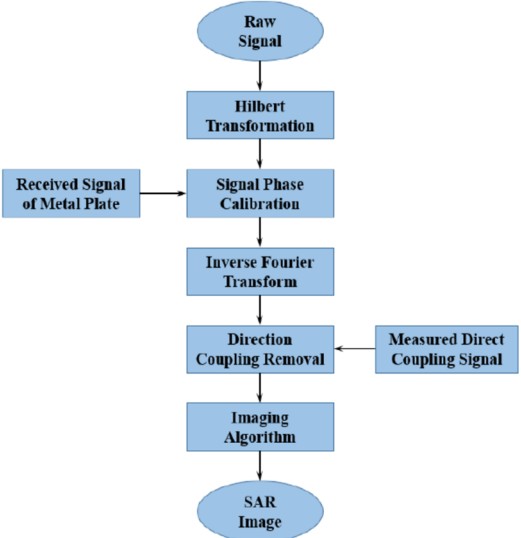

**Figure 17.** Signal processing flowchart for the designed MIMO radar system.

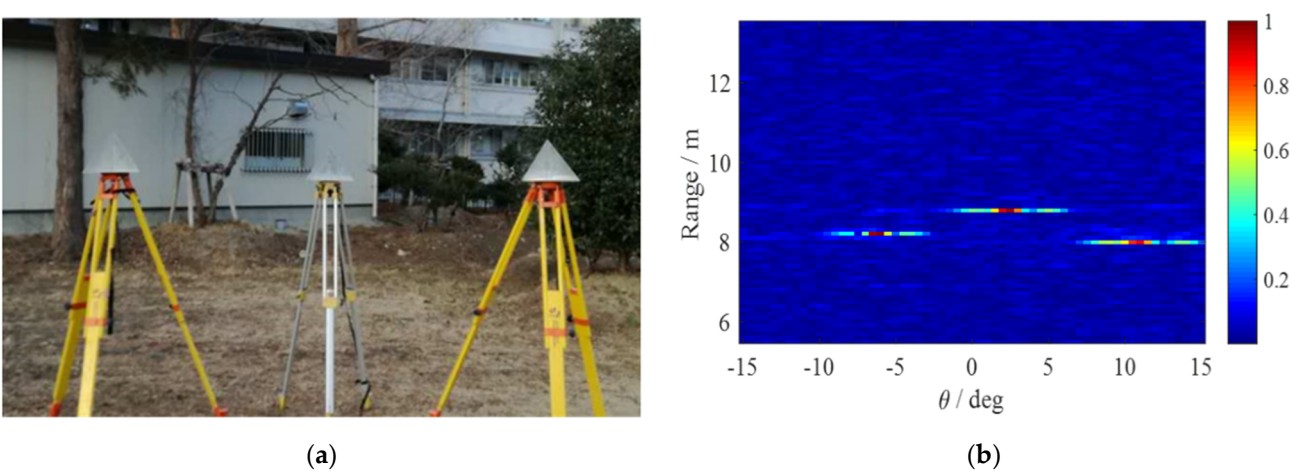

(**a**)            (**b**)

**Figure 18.** (**a**) Three trihedral CRs used as targets. (**b**) SAR image of the CRs obtained by the designed MIMO radar system.

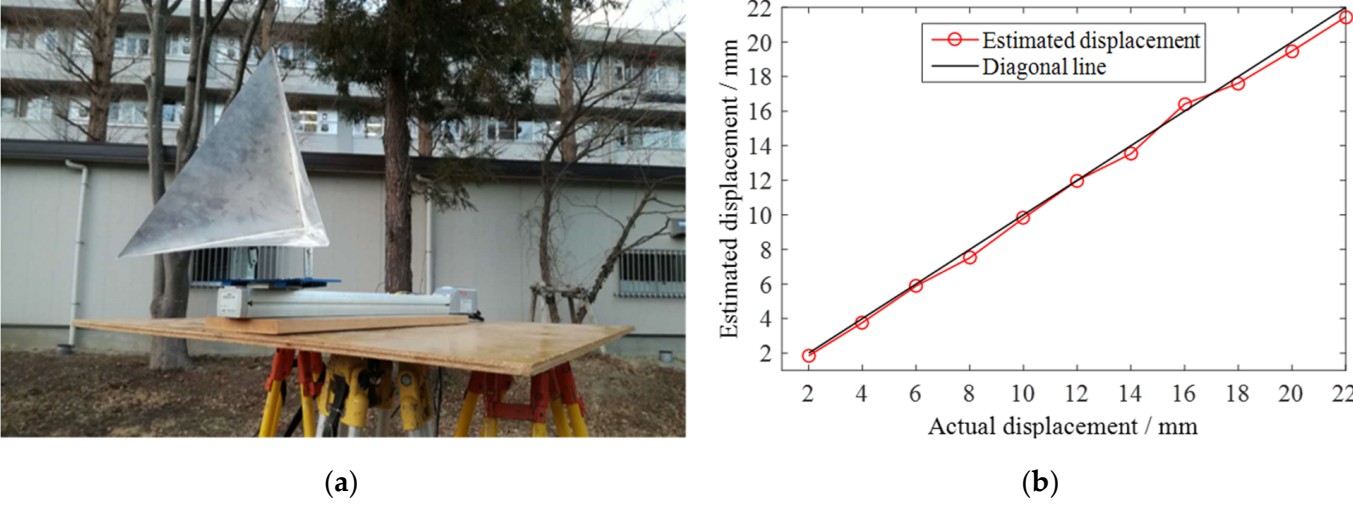

(**a**)                              (**b**)

**Figure 19.** (**a**) Trihedral CR moved on a linear scanner. (**b**) Displacement estimation of the CR obtained by the designed MIMO radar system.

Using the designed MIMO radar system, and in a period of 40 s, we measured the displacement of a bridge resulting from the passing of a truck. The bridge is shown in Figure 20. We can see that the bridge has four horizontal and five vertical bars, which can be deformed by the heavy truck and thus are the principal areas for monitoring during this experiment. A SAR image of the bridge, obtained by the BP algorithm, is shown in Figure 21. It is evident that the structure of the bridge can be well depicted by the designed radar system. Further, the pier of the bridge can also be imaged, as indicated by the rectangles in Figures 20 and 21. The monitoring findings closely aligned with outcomes observed using linear variable differential transformer (LVDT) sensors in the context of urban railroad bridges [51].

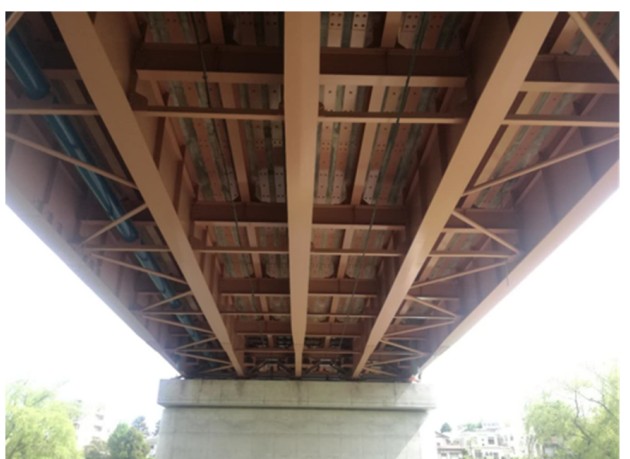 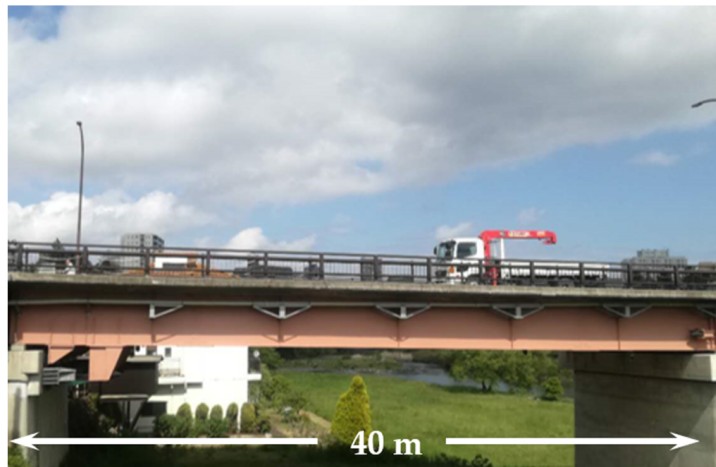

**Figure 20.** Concrete road bridge under measurement.

Thereafter, differential interferometric processing was conducted to estimate the displacement of four different points of the bridge. These four selected points had the highest amplitudes, which resulted in acceptable estimation precision, and different ranges and angles from which to verify the 2D separation capacity of the designed MIMO radar system. The displacement estimation results are also shown in Figure 21, from which it can be observed that these four points experienced different displacements along the LOS. Moreover, different from the RAR measurement shown in the above section (which can provide good range resolution only and thus loses information in the azimuth direction), the 2D

imaging capacity and fast data sampling rate of the designed MIMO radar system make separation of the targets in both the range and azimuth directions possible. Thus, more accurate displacement estimation can be achieved.

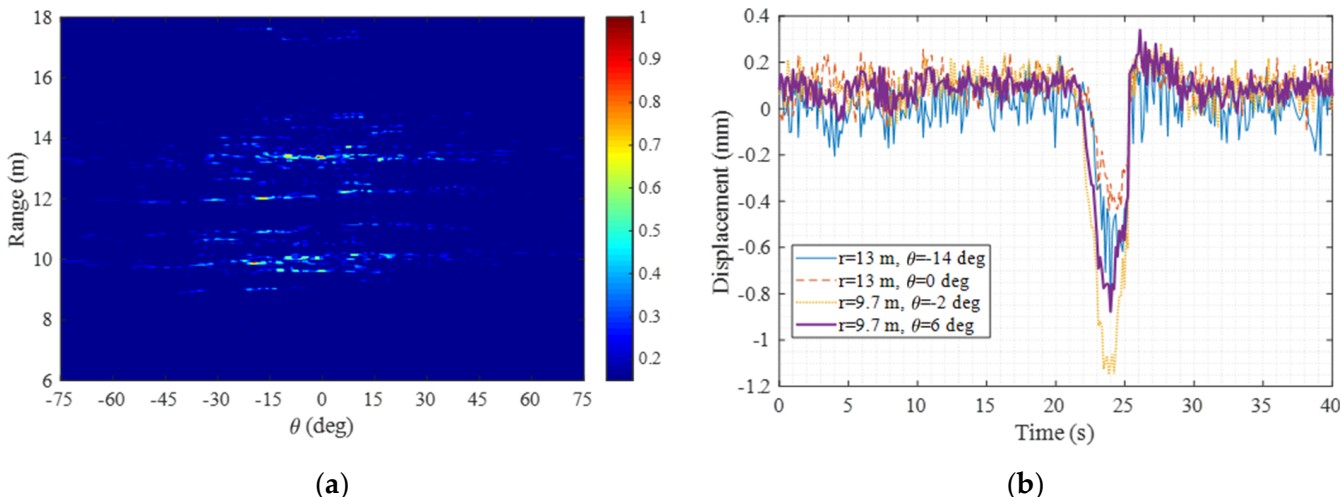

(**a**)                                                                            (**b**)

**Figure 21.** (**a**) SAR image of the bridge obtained by the BP algorithm. (**b**) Displacements of four different points of the bridge.

## 4. Conclusions

Ground-based microwave radar interferometry stands at the forefront of recent innovations in infrastructure health monitoring. This technique offers the unique advantage of providing comprehensive insights into the tested structure's global behavior. However, a notable drawback is the challenge of precisely localizing measured displacements on the structure due to the contribution of all points within the same resolution cell.

This paper introduced innovative strategies to overcome the challenges associated with ground-based microwave radar interferometry in bridge monitoring. The study showcased monitoring results from various bridges using different radar systems. A long-span metallic railway bridge was monitored under a polarimetric real aperture radar system, while a stone bridge and a concrete road bridge underwent measurement via a synthetic aperture radar system, employing a linear rail and MIMO array.

Novel signal processing methodologies were proposed, emphasizing their capability to provide precise location and vertical deformation in structure measurements. The advantages of the microwave radar system—specifically the MIMO array design methodology—and its application to bridge monitoring were elucidated. The MIMO radar system offers not only full-range and cross-range resolution but also an exceptionally high acquisition rate, catering to both dynamic and static monitoring purposes.

The research delved into the dynamic responses of three bridge types—a passing metallic bridge, a stone bridge, and a concrete bridge—under loading conditions. The findings underscore the capability of the microwave radar interferometer, coupled with advanced algorithms, to effectively monitor any bridge type with unprecedented spatial and temporal resolution. This breakthrough marks a significant advancement in bridge health monitoring, promising enhanced accuracy and efficiency through radar technology.

In summary, ground-based microwave radar interferometry emerges as a powerful tool for assessing infrastructure health, offering a holistic view of a structure's behavior. Despite challenges in precise localization, this paper presented inventive strategies and showcased successful monitoring results on various bridges. The proposed signal processing methodologies and the application of the MIMO radar system further demonstrated the potential to revolutionize bridge monitoring, setting a new standard for spatial and temporal resolution in the field.

**Author Contributions:** Conceptualization, L.Z. and W.F.; methodology, L.Z., W.F., O.M. and G.N.; software, L.Z., W.F. and O.M.; validation, L.Z., W.F., O.M., G.N. and M.S.; formal analysis, L.Z., W.F., O.M. and G.N.; investigation, L.Z., W.F., O.M. and G.N.; resources, W.F., G.N. and M.S.; data curation, L.Z., W.F. and G.N.; writing—original draft preparation, L.Z., W.F. and G.N.; writing—review and editing, G.N., A.M.A. and M.S.; visualization, G.N., A.M.A. and M.S.; supervision, G.N., A.M.A. and M.S.; project administration, G.N. and M.S.; funding acquisition, G.N., A.M.A. and M.S. All authors have read and agreed to the published version of the manuscript.

**Funding:** This research received no external funding.

**Institutional Review Board Statement:** Not applicable.

**Informed Consent Statement:** Not applicable.

**Data Availability Statement:** The data presented in this study are available on request from the corresponding author.

**Conflicts of Interest:** Author Olimpia Masci was employed by the company DIAN s.r.l. The remaining authors declare that the research was conducted in the absence of any commercial or financial relationships that could be construed as a potential conflict of interest.

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
