# Peer review of "Bridge Monitoring Strategies for Sustainable Development with Microwave Radar Interferometry"

_sustainability, doi:10.3390/su16072607_

Round 1
Reviewer 1 Report
Comments and Suggestions for Authors
The manuscript presents microwave radar systems to handle the dynamic responses of bridges under different loading conditions, which is an important topic on monitoring technologies for sustainable infrastructures. The manuscript is easy to follow but some issues need to be solved or clarified.
(1) The introduction section is too sketchy. The backgrounds and necessary related literature has not been investigated in depth. The uniqueness or novelty of this manuscript should be more highlighted and the authors are encouraged to compare with the existing monitoring methods using different sensors. In this sense, the authors need to rearrange the section in 4-5 paragraphs, so that readers can understand why and how the authors have proposed new findings or new methodology.
(2) The authors need to add more notations in the figures such as Figs. 1, 6, 11, 16, 19, and these figures will become more self- explanatory, e.g., the lengths and dimensions can be added into the bridge in Fig. 6.
(3) Is there any equivalent evaluation indicator for the measurements? For instance, how to judge the polarimetric responses (e.g. Fig. 3 and Fig. 17). This concern should be addressed. In addition, the reviewer suggests the authors to add a flowchart with some pictures in the beginning of Section 2, linking equipment setup, data acquisition and processing, result evaluation, etc. depending on loads or environments.
(4) The reviewer suggests the authors to re-arrange the concluding part in a point-by-point manner, and only present the novelty and essential findings. As there are many data involved, the machine learning techniques, e.g. 10.1016/j.jrmge.2022.10.014, can be helpfully discussed.
Comments on the Quality of English LanguageNeed minor editing of language.
Author Response
The authors thank the editor for allowing us with an opportunity to revise the manuscript. We appreciate the time and effort the reviewers spent in reading our manuscript and providing constructive comments, which have helped improve the quality of our revised manuscript. In this revision, all the comments from the three reviewers have been addressed. We have outlined below our responses to reviewers’ comments point by point.

Reviewer 2 Report
Comments and Suggestions for Authors
The presented paper explores the possibility of gathering information about the oscillations induced on a bridge by passing vehicles. The proposed solutions aim to enhance measurement accuracy and facilitate the study of potential construction risks.
I would recommend considering the correlation of these results with the health of the structure, particularly in terms of fatigue that could lead to catastrophic failure.
Addressing the issue of measurement accuracy is paramount. Could these results be utilized to install measuring devices beneath sensitive structures? The authors should discuss the feasibility of implementing continuous structural degradation monitoring or conducting periodic measurements to assess if any detected changes correspond to possible defect development. Additionally, it is worth exploring whether these measurements can identify defects resulting from severe weather events, such as strong winds, storms, and extreme temperature fluctuations associated with climate change.
I suggest that the authors incorporate these topics into both the introduction and discussion sections to provide a more comprehensive examination of the implications of their research.
Author Response

(The authors gave the same response as above.)

Reviewer 3 Report
Comments and Suggestions for Authors
This is an interesting paper, however, some major aspects need to be addressed before publication:
- Firstly, the authors should compare the measurements obtained with the radar to those from conventional sensors, such as contact sensors, in each case. This means comparing the measurements from the radar with those from alternative measurement devices, and analyzing the error relative to the high precision and direct ones.
- Please provide details on the measurement range, resolution, and sensitivity of this measurement device. Note that in bridges, the vertical deflection can be very small, often in the order of hundredths of a millimeter, as suggested in the references. This level of resolution, often not attainable even with contact sensors, is necessary.
- Measuring vertical deflection is a complex task and may only be feasible in a few cases, especially with long-span bridges. Please provide error metrics as a function of the average deflection value. Is the hypothesis of homoscedasticity valid in this context?
- Regarding the considered MIMO system, what is the synchronization error among measurements? Discuss whether this error is compatible with operational modal analysis post-processing, and if so, provide a demonstration.
Suggested references for inclusion:
- Aloisio, A., Contento, A., Alaggio, R., & Quaranta, G. (2023). Physics-based models, surrogate models, and experimental assessment of the vehicle–bridge interaction in braking conditions. Mechanical Systems and Signal Processing, 194, 110276.
- Tang, Y., Cang, J., Zheng, B., & Tang, W. (2023). Deflection Monitoring Method for Simply Supported Girder Bridges Using Strain Response under Traffic Loads. Buildings, 14(1), 70.
Author Response

(The authors gave the same response as above.)

Round 2
Reviewer 1 Report
Comments and Suggestions for Authors
The paper has been much improved and the reviewer would like to recommend publication.
Author Response
The authors thank the editor for allowing us with an opportunity to revise the manuscript. We appreciate the time and effort the reviewers spent in reading our manuscript and providing constructive comments, which have helped improve the quality of our revised manuscript.
Reviewer 3 Report
Comments and Suggestions for Authors
Accept
Author Response

(The authors gave the same response as above.)
